# Transcriptomic-Metabolomic Profiling in Mouse Lung Tissues Reveals Sex- and Strain-Based Differences

**DOI:** 10.3390/metabo12100932

**Published:** 2022-09-30

**Authors:** Jolyn Fernandes, Katelyn Dunigan-Russell, Hua Zhong, Vivian Lin, Mary Silverberg, Stephanie B. Moore, ViLinh Tran, Dean P. Jones, Peter F. Vitiello, Lynette K. Rogers, Trent E. Tipple

**Affiliations:** 1Section of Neonatal-Perinatal Medicine, Department of Pediatrics, College of Medicine, University of Oklahoma Health Sciences Center, Oklahoma City, OK 73104, USA; 2Pulmonary, Critical Care and Sleep Medicine, Davis Heart and Lung Research Institute, Ohio State University Wexner Medical Center, Columbus, OH 43210, USA; 3Neonatology, Department of Pediatrics, University of Alabama at Birmingham, Birmingham, AL 35294, USA; 4Clinical Biomarkers Laboratory, Department of Medicine, Emory University, Atlanta, GA 30322, USA; 5Center for Perinatal Research, Nationwide Children’s Hospital, Columbus, OH 43215, USA; 6Department of Pediatrics, Ohio State University, Columbus, OH 43210, USA

**Keywords:** metabolomics, transcriptomics, xMWAS, sex differences, strain differences

## Abstract

Omics analyses are commonly used for identifying pathways and genes responsible for physiologic and pathologic processes. Though sex is considered a biological variable in rigorous assessments of pulmonary responses to oxidant exposures, the contribution of the murine strain is largely ignored. This study utilized an unbiased integrated assessment of high-resolution metabolomic profiling and RNA-sequencing to explore sex- and strain-dependent pathways in adult mouse lungs. The results indicated that strain exhibited a greater influence than sex on pathways associated with inflammatory and oxidant/antioxidant responses and that interaction metabolites more closely resembled those identified as differentially expressed by strain. Metabolite analyses revealed that the components of the glutathione antioxidant pathway were different between strains, specifically in the formation of mixed disulfides. Additionally, selenium metabolites such as selenohomocystiene and selenocystathionine were similarly differentially expressed. Transcriptomic analysis revealed similar findings, as evidenced by differences in glutathione peroxidase, peroxiredoxin, and the inflammatory transcription factors RelA and Jun. In summary, an multi-omics integrated approach identified that murine strain disproportionately impacts baseline expression of antioxidant systems in lung tissues. We speculate that strain-dependent differences contribute to discrepant pulmonary responses in preclincal mouse models, with deleterious effects on clinical translation.

## 1. Introduction

Omics analyses have become state-of-the-art for identifying pathways and genes responsible for physiologic and pathologic processes. Transcriptomic investigations in mouse lungs have revealed distinct patterns of differentially expressed genes in both sexes in response to newborn hyperoxia exposure, specifically in antioxidant pathways [1,2]. Others have noted substantial differences between inbred mouse strains [2,3,4,5,6] and interactions between sex and strain in the context of oxidant/antioxidant reactions to oxidant stress, specifically hyperoxia exposure [2,6,7,8,9]. While the identification of alternatively expressed genes and proteins have contributed significantly to our understanding of numerous pulmonary pathologies, metabolomic analyses have gained wide-spread acceptance as a novel platform that can provide functional information on biochemical pathways downstream of transcripts and proteins. For this reason, metabolomics has added value for understanding stress responses to toxic exposures [10]. At present, there is a paucity of unbiased information on baseline differences in metabolite profiles between mouse strains and sexes.

Oxygen therapy is used to treat pulmonary insufficiency in premature infants and critically ill patients with downstream effects on lung development and function. Our lab has focused on the mechanisms of hyperoxic lung injury in newborn and adult mouse models, and we and others have identified altered expression of inflammatory and oxidant/antioxidant genes [5,11,12,13]. We have previously reported that C3H/HeN and C57Bl/6N mice and male and female mice exhibit differential susceptibilities to hyperoxia [5,12,14,15]. Furthermore, we have identified significant differences in nuclear factor E2-related factor (Nrf2, an antioxidant transcription factor) activity between these strains, which led to differential responses to therapeutic interventions [5]. Furthermore, the National Institutes of Health and similar agencies consider sex as a biological variable to be a prerequisite for scientific rigor. Collectively, these observations prompted the present study investigating differential pulmonary expression of inflammatory, oxidant, and antioxidant pathways utilizing an integrated multi-omics approach.

The current investigation was designed to explore strain-based (C3H/HeN vs. C57Bl/6N) differences, sex-based (male vs. female) differences, and sex/strain-based interactions in adult lungs using a combination of metabolomic profiling integrated with RNA-sequencing. Given our previously published findings in these two strains and our interests in understanding strain-based differences in response to oxidant stress, we chose unbiased approaches to define baseline inflammatory and antioxidant genes in the absence of exogenous stimulus. A source-matched sample cohort (different samples from the same biological organism) was obtained for omics analyses, and the data were analyzed using both conceptual and pathway-based integration platforms [16]. Our goal was to identify unperturbed differences within each strain and sex that might influence the differential pulmonary responses to oxidants, and environmental and/or xenobiotic exposures.

## 2. Materials and Methods

### 2.1. Mice

C57Bl/6N and C3H/HeN adult mice (6–8 weeks old) were purchased from Harlan (Indianapolis, IN). After acclimation, the mice were euthanized and lung tissues were snap-frozen and stored at −80 °C until needed. A total of 12 mouse lungs were analyzed, comprising equal numbers of each strain, C57Bl/6N (C57B) and C3H/HeN (C3H), and sexes (males (M) and females (F)). As such, we had n = 3 for each of the four groups, including C3HM, C57BM, C3HF, and C57BF. Animal studies were performed at The University of Alabama Birmingham using protocols approved by the Institutional Animal Care and Use Committee (IACUC #20001). Additionally, studies were conducted in accordance with ARRIVE guidelines.

### 2.2. High-Resolution Metabolomics

For metabolomics, the samples were prepared as previously described by Chandler et al. [17] Twenty to thirty milligrams of lung tissue were extracted by homogenization in 15 µL/mg of 2:1 acetonitrile:water containing a mixture of stable isotope-labeled internal standards [18]. The samples were incubated on ice for 30 min and then centrifuged to remove precipitates. The supernatants were stored at 80 °C until LC/MS analysis [17]. High-resolution mass spectrometry with liquid chromatography was used to collect metabolic data of lungs from mice [19]. Briefly, the samples were analyzed with three technical replicates using hydrophilic interaction liquid chromatography (HILIC) (Waters Xbridge BEH amide column) on an Orbitrap Fusion Tribrid Mass Spectrometer (Thermo Scientific, Waltham, MA, USA). Flow rate was maintained at 0.35 mL/min until 1.5 min, increased to 0.4 mL/min at 4 min, and held for 1 min. Solvent A was 100% LC-MS grade water, solvent B was 100% LC-MS grade acetonitrile, and solvent C was 2% formic acid (*v/v*) in LC-MS grade water. The initial mobile phase conditions were 22.5% A, 75% B, and 2.5% C, with linear gradient set to 77.5% A, 20% B, and 2.5% C, resulting in a total analytical run time of 5 min. The instrument conditions were spray voltage, 3500 (V); capillary temperature, 300 °C; sheath gas flow, 45 (arbitrary units); auxillary gas flow, 25 (arbitrary units); spare gas flow, 1 (arbitrary units); max spray current, 100 (μA); probe heater temperature, 200 °C. Data extraction and filtration were performed by apLCMS [20] and xMSanalyzer [21].

There were 19,559 metabolite features extracted for downstream statistical analysis. Data were filtered for strain and sex resulting in 15,532 metabolites. The LIMMA two-way ANOVA, LIMMA decide test, and partial least squares discriminant analysis (PLS-DA) were implemented by R version 4.1.0 with package xmsPANDA version 1.3.2 (https://github.com/kuppal2/xmsPANDA) to select significant features by either sex, strain, or interaction. Data filtration criteria included replacing missing values by half feature minimum value to include data with at least 70% values for each technical replicate, 50% values throughout each feature, and 50% values for each group. The significant threshold was set at *p* < 0.05, with the correction for false discovery rate (FDR) at *q* < 0.05 using Benjamini and Hochberg (BH) correction method [22]. For annotation of metabolites, xMSannotator [23] was used based upon Human Metabolomics DataBase (HMDB) and Kyoto Encyclopedia of Genes and Genomes [24] with 5 ppm tolerance. To investigate and identify the principal metabolites associated with each factor across all groups, an unsupervised two-way hierarchical clustering analysis (HCA) was performed. To perform the metabolomic enrichment analysis, all significant metabolite features by each factor were analyzed in Python version 3.9.6 with package mummichog version 2.4.4 [19]. Significant pathways were selected using cut off value at *p* < 0.05.

### 2.3. Transcriptomics by RNA-Sequencing

Transcriptomics analyses were performed by Genewiz, Inc. (South Plainfield, NJ, USA) using the following protocol. Library Preparation with polyA selection and HiSeq Sequencing RNA samples received were quantified using Qubit 2.0 Fluorometer (Life Technologies, Carlsbad, CA, USA), and RNA integrity was checked using Agilent TapeStation 4200 (Agilent Technologies, Palo Alto, CA, USA). RNA sequencing libraries were prepared using the NEBNext Ultra RNA Library Prep Kit for Illumina following manufacturer’s instructions (NEB, Ipswich, MA, USA). Briefly, mRNAs were first enriched with Oligo(dT) beads. Enriched mRNAs were fragmented for 15 min at 94 °C. First-strand and second-strand cDNAs were subsequently synthesized. cDNA fragments were end-repaired and adenylated at 3′ends, and universal adapters were ligated to cDNA fragments, followed by index addition and library enrichment by limited-cycle PCR. The sequencing libraries were validated on the Agilent TapeStation (Agilent Technologies, Palo Alto, CA, USA), and quantified by using Qubit 2.0 Fluorometer (Invitrogen, Carlsbad, CA, USA) as well as by quantitative PCR (KAPA Biosystems, Wilmington, MA, USA). The sequencing libraries were clustered on lanes of a flowcell. After clustering, the flowcell was loaded on the Illumina HiSeq instrument (4000 or equivalent) according to manufacturer’s instructions. The samples were sequenced using a 2 × 150 bp Paired End (PE) configuration. Image analysis and base calling were conducted by the HiSeq Control Software (HCS). Raw sequence data (.bcl files) generated from Illumina HiSeq was converted into fastq files and de-multiplexed using Illumina’s bcl2fastq 2.17 software. One mismatch was allowed for index sequence identification. Derived sequences were processed by applying a custom computational pipeline [25], wherein reads generated were mapped to the mice genome (mm10) using open source gSNAP [26] and expression (FPKM) derived by Cufflinks [27].

There were 47,561 transcripts annotated with an official gene symbol and used in this study. Pre-processing and data normalization resulted in 18,544 transcripts that were used for further analysis. To select significant transcriptome features by either gender or strain or interaction, the LIMMA two-way ANOVA, LIMMA decide test, and PLS-DA were conducted by R version 4.1.0. To investigate the effect of each factor, a Gene Set Enrichment Analysis, GSEA version 4.1.0 [28] was used, wherein all transcripts selected by each factor were pre-ranked by their logarithm fold change (log_2_FC). Sets containing 15–500 genes were analyzed, wherein the dataset was collapsed using Molecular Signatures Database (MSigDB) with hallmark gene set version 7.4 [29].

### 2.4. Transcriptome–Metabolome Wide Association Study (TMWAS)

For the interaction between metabolites and transcriptomics, the following significant metabolic features—strain (921), gender (660), and interaction (374)—were used for integration with the whole 18,544 transcriptomic features using xMWAS version 0.56 [30]. Oxidant/antioxidant metabolites and enriched transcripts associated with antioxidant and inflammatory pathways were analyzed for significant correlations. In this integration analysis, we used sparse partial least squares (sPLS) in canonical mode to conduct pairwise correlation analysis. The pairwise integrative analysis included the correlation threshold at 0.9, and the significant correlations were selected at *p*-value threshold 0.05. Selected metabolic features were mapped to metabolic pathways using mummichog and annotated with xMSannotator [23] using KEGG database [31]. For ease of presentation, corresponding clusters or pathways between metabolomics and transcriptomics were defined as communities.

## 3. Results

### 3.1. Differences in Metabolome

There were 921 differentially expressed metabolites between strains, 660 between sexes, and 374 at the interaction between sex and strain (Appendix A). The results of a two-way hierarchical cluster analysis are presented as heatmaps (Figure 1A–C). For strain, (Figure 1A), the results indicated two main clusters of metabolic features that differed between C3H and C57B strains. Between C3H and C57B, 507 features were increased and 414 features were decreased in C3H compared to C57B. For sex (Figure 1B), the results indicated two main clusters for male and female comparisons. Between male and female, 319 features were increased, and 341 features were decreased. The interaction results (Figure 1C) indicated four main clusters—C3HM, C57BM, C3HF, and C57BF—with the metabolic features being partially different among these four clusters. The results of the interactions from the PLS-DA analysis are presented as score plots (Figure 1D). The separation according to interactions within component 1 is 77.84% and that for component 2 is 22.16%. Pathway enrichment analyses (Appendix A) with metabolites among strain, sex, and interactions are presented as a bubble plot (Figure 1E). The bubble plot depicts metabolic pathways identified as different based on statistically relevant absolute differences in metabolite content. For metabolomics, the bubble size corresponds to the absolute expression of the log value of a given group of metabolites (*p* < 0.05). Across strain, sex, and interaction, the unique and common metabolic pathway are displayed by Venn diagram in (Appendix A).

For this investigation, significant primary metabolic pathways associated with inflammation, antioxidants, and oxidant stress were the focus of further analyses (Figure 2). Metabolites within the methionine, cysteine, and selenocysteine amino acid pathways were of particular interest due to antioxidant functions. Selected metabolites for strain, sex, and interaction are provided in the plots displayed in Figure 2A–C. For strain (Figure 2A), L-cysteinylglycine and selenohomocystiene were lower in C3H than C57B. For sex (Figure 2B), selenocystathionine and S-prenyl-L-cysteine were higher in females than males. The interactions between strain and sex identified oxidized products of the cysteine/glutathione pathways and included glutathione disulfide, cysteine-glutathione disulfide, L-cysteinylglycine disulfide, and homocysteine sulfinic acid (Figure 2C). Glutathione disulfide was higher in the C3H strain and C57BF than in C57BM. Cysteineglutathione disulfide was lower in the C3H than in C57B of both sexes. L-cysteinylglycine disulfide was lower in C3H than C57B strains. Homocysteine sulfinic acid was high in C3HF and low in C3HM, with both sexes of C57B at an intermediate level.

### 3.2. Differences in Transcriptome

Transcriptomic data yielded differentially expressed transcripts: 1418 between strains, 484 between sexes, and 383 at the interaction between sex and strain (Appendix A). The results of the two-way hierarchical cluster analyses are presented using heatmaps (Figure 3). Strain (Figure 3A) resulted in two main clusters for C3H and C57B. Expression was different between strain C3H and C57B, with 703 genes higher and 715 genes lower in C3H than C57B. Sex (Figure 3B) resulted in two main clusters between males and females. The transcriptome analyses yielded differences between males and females, with 196 genes higher and 288 genes lower in males than females. The interactions (Figure 3C) yielded four main clusters: C3HM, C3HF, C57BM, and C57BF. The PLS-DA score plots show that separation according to interactions in component 1 is 59.34% and that for component 2 is 40.66% (Figure 3D). The gene set enrichment analysis (Appendix A) of transcripts among strain, sex, and interaction are presented using bubble plots (Figure 3E). The size of the bubble correlates with the size of the absolute enrichment score (ES), which represents both increased and decreased expression and uses an absolute score of change for statistical analyses. Across strain, sex, and interaction, the unique and common transcriptomic pathways are displayed in a Venn diagram (Appendix A).

An analysis of the primary transcriptomic pathways identified several differentially expressed transcripts that may be significant in the context of inflammation and oxidant stress (Figure 4). Select transcripts for strain, sex, and interaction are provided in plots displayed in Figure 4A–C. For strain, glutathione peroxidase 2 and glutamate cysteine ligase C were lower in C3H than C57B (Figure 4A). For sex, Rela (which encodes p65, an NFkB subunit) and Jun (an activator protein-1, AP-1, subunit) were higher in females than males (Figure 4B). Interactions between strain and sex included transcripts for antioxidants, specifically catalase, peroxiredoxin 1, selenoprotein P, and the oxidant producing enzyme nitric oxide synthase 2. Catalase was lower in C3HM and both sexes of C57B than in C3HF. Peroxiredoxin 1 was only modestly different with C57BF lower than C57BM and female and male C3H. Selenoprotein P was lower in C3HM and C57BF than the respective males. Nitric oxide synthase 2 (iNOS) was lower in C3HM and both sexes of C57B than in C3HF (Figure 4C).

### 3.3. Integrated Transcriptome-Metabolome Wide Association Study

As a complementary approach to analyzing individual omics data sets, the data were analyzed for significant differences in strain, sex, and interaction using xMWAS. This approach provides interaction information between the two omics layers in an unbiased correlation model to detect distinct communities (clusters or pathways) with functionally relevant connection nodes [30,32]. First, select metabolites associated with oxidant/antioxidant pathways within specific communities were identified, and associated transcripts were searched using enriched data sets. Only communities that contained metabolites/genes associated with oxidant/antioxidant or inflammatory pathways were annotated. The greatest numbers of differentially expressed selected metabolites were identified within strain, and an analysis of transcripts yielded several genes associated with antioxidants and inflammatory responses (Figure 5 and Appendix A). Sex and interaction analyses yielded fewer differentially expressed oxidant/antioxidant-related metabolites, but those identified were associated with many of the same strain-associated inflammatory genes. Significantly, Communities 3 and 4 contained differentially expressed metabolites and genes of interest in all three groups: strain, sex, and interaction.

Similarly, genes associated with inflammation and antioxidant pathways within specific communities were identified, and associated metabolites were searched from enriched data sets (Figure 6 and Appendix A). Both strain and sex yielded significant numbers of differentially expressed metabolites associated with the chosen transcripts within several communities. Many of the metabolites were members of the cysteine/glutathione or selenium-associated antioxidant pathways.

## 4. Discussion

Mouse models utilizing C57Bl/6 mice are frequently used due to the number of transgenic and knockout mice created in this background. Investigations in non-C57Bl/6 mice have revealed substantial inter-strain differences with or without external stressors. Nichols et al. reported a comparison of newborn responses to hyperoxia exposure in 36 inbred stains using GWAS and identified differentially coded QTLs. Their data indicated that genetic backgrounds are an important determinant in inflammatory responses and identified candidate susceptibility genes [3]. Using RNA seq, Coarfa et al. investigated neonatal male versus female differences in hyperoxic responses using the C57Bl/6N strain [1]. Distinct transcriptomic responses were identified using gene set enrichment analysis and pathways that were regulated in the opposite directions in controls and hyperoxia-exposed mice. Notably, several redox regulated transcription factors were included in the differentially expressed genes including *Nrf2*, *Ap1*, *NFkB* and *HIF-1α*. Leary et al. investigated the effects of an outbred strain versus an inbred strain and sex in a newborn mouse model of hyperoxic lung injury. While sex-dependent differences were noted, the effect of strain on lung growth parameters was greater. Others have evaluated effects, specifically in the context of hyperoxia between C57BL/6J, BALB/cJ, FVB/NJ, C3H/HeJ, and DBA/2J strains and have reported substantial differences in antioxidant pathway responses [4]. Baseline differences in the expression of inflammatory and oxidant/antioxidant pathways are likely responsible for strain-dependent differential responses to stressors including hyperoxia exposure.

Transcriptomics by RNA-seq is now used extensively in most research environments but provides limited information on the products of the transcripts and the subsequent metabolites, which more directly influence physiology. As a newer area of investigation, metabolomics has considerable potential and provides information in a more organ or pathway specific manner. The integration of transcriptomics and metabolomics is more challenging than either analysis alone as the two are not direct results of one another but, rather, reflect a “cause and subsequent effect” relationship. Consequently, the present investigation was focused on lung profiles of healthy, non-exposed mice, and an untargeted approach was used to facilitate identification of metabolome/transcriptome interactions, specifically those responsible for inflammatory and oxidant/antioxidant pathway activation.

The results of our non-targeted metabolomics approach yielded strain, sex, and interaction differences between C3H/HeN and C57Bl/6N mice and male and female sex (Figure 1). There was minimal overlap in the significant pathways identified as indicated in Figure 1D,E. When focusing on inflammation, oxidant, and antioxidant metabolites, we observed differential expression of many metabolites in the methionine/cysteine/glutathione pathways. C3H/HeN lung tissues had significantly more oxidized glutathione (glutathione disulfide) than C57Bl/6N lung tissues; however, other mixed oxidation products such as cysteineglutathione disulfide and L-cysteinylglycine disulfide were higher in the C57Bl/6N than C3H/HeN lung tissues (Figure 2). One possible explanation is that C3H/HeN mice have a more robust lung glutathione system (synthesis or reserve) and are better able to regulate oxidation to prevent the formation of mixed disulfides with other thiol-containing compounds [13].

Substantially more overlap between strain, sex, and interaction was observed in our transcriptional analyses (Figure 3E). Many of the differentially expressed pathways were related to inflammation, specifically, TNFα signaling via NFκB, inflammatory response, interferon gamma response, and p53 pathways. Mice of the same strain tended to group together, regardless of sex. We interpret this finding as indicating that sex has less effect on gene expression than does strain within these comparisons (Figure 3D). The identification of inflammation, oxidant, and antioxidant transcripts indicated the presence of differential expression of many antioxidant genes including glutathione peroxidase 1, catalase and peroxiredoxin 1, in C3H/HeN compared to C57Bl/6N lung tissues (Figure 4).

To identify true correlations between metabolites and transcripts, data were alternatively analyzed in two distinct ways. First, metabolites relevant to oxidant/antioxidant functions were chosen. Using only enriched metabolites, correlations between oxidant/antioxidant metabolites and transcripts associated with antioxidant or inflammatory pathways (which may be associated with oxidation) were subsequently identified (Figure 5). Differential expression between strains yielded the most significant correlations between metabolites of the cysteine/glutathione pathway and transcripts associated with metabolic function. Again, sex-based analyses yielded substantially fewer correlations and the interaction group more resembled strain than sex with fewer communities identified as containing metabolites of interest.

As a second analysis, transcripts for antioxidant and inflammatory genes were chosen, and correlations between the chosen transcripts and oxidant/antioxidant metabolites were analyzed (Figure 6). Strain yielded the largest number of differentially expressed transcript–metabolite correlations and primarily correlated with cysteine/glutathione or selenium antioxidant metabolites. Correlations between differentially expressed transcripts and metabolites were fewer between sexes and included antioxidant genes and metabolites associated with cysteine/glutathione, methionine, or selenium antioxidants. The correlations in the interaction group were substantially fewer and were largely antioxidant genes and oxidation product metabolites.

The analysis of only two strains of mice is an acknowledged limitation of this investigation, although we intentionally focused on the strains given in our previously published findings of differential antioxidant response to hyperoxia [5]. Similarly, our focus on inflammation, oxidation, and antioxidants represents another limitation that is justified by our intent to explore pathways with relevance to hyperoxic lung injury that align with our ongoing interests. We readily acknowledge that there are much data left unexplored and that this dataset offers a rich source of new and potentially important information on other areas of research. The data presented provide additional support for the need to acknowledge differential expression of genes and metabolites in different mouse strains when designing rigorous preclinical studies, including those investigating exposure and/or therapeutic interventions. The finding that strain had a greater influence than sex on inflammation, oxidation, and antioxidant pathways is especially notable in light of increased focus on sex as a biological variable in scientific rigor and reproducibility [33,34]. In summary, we have incorporated a multi-omics integrated approach and have identified disproportionate differences between stain and sex at baseline in the murine lung. These data will be used to provide a basis for identifying the underlying causes to discrepant responses in different mouse models.

## Figures and Tables

**Figure 1 metabolites-12-00932-f001:**
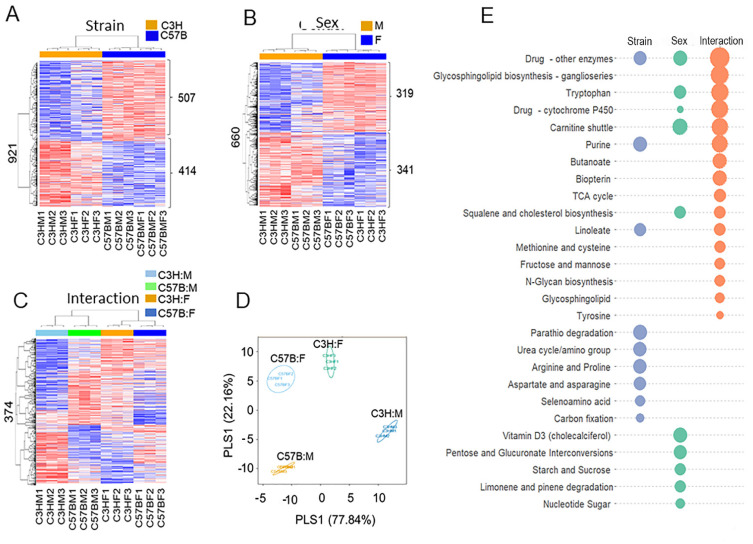
Differentially expressed metabolites. The results of two-way hierarchical clustering presented as heatmaps for (**A**) strain, (**B**) sex, and (**C**) interactions. (**D**) PLS-DA score plot for high-resolution metabolomics data resulting in strain, sex, and interaction. (**E**) Pathway enrichment analyses among strain, sex, and interactions as a bubble plot with bubble size corresponding to the size of the log value (*p* < 0.05).

**Figure 2 metabolites-12-00932-f002:**
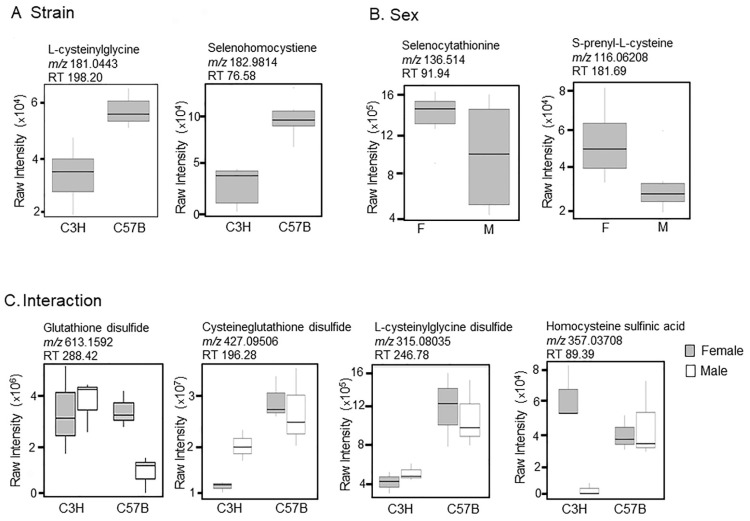
Metabolites identified within the methionine, cysteine, and selenocysteine pathways. Selected metabolites which differed between (**A**) strain, (**B**) sex, and (**C**) interaction were plotted.

**Figure 3 metabolites-12-00932-f003:**
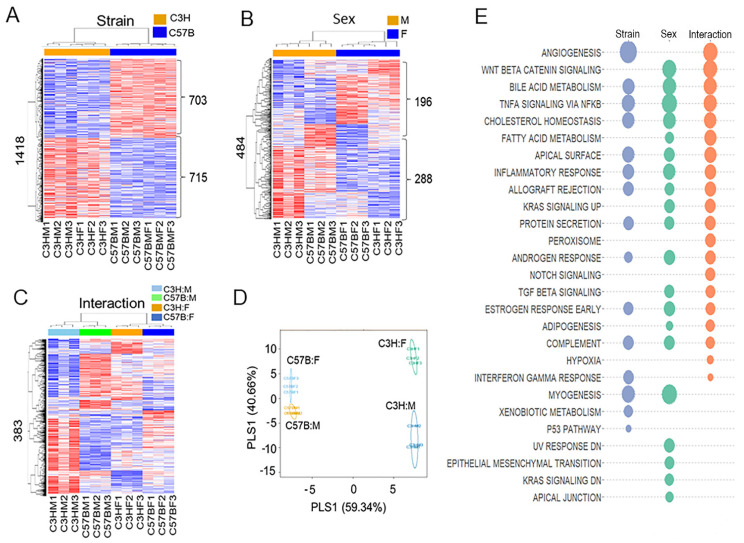
Differentially expressed transcripts. The results of two-way hierarchical clustering presented as heatmaps for (**A**) strain, (**B**) sex, and (**C**) interactions. (**D**) PLS-DA score plot for high-resolution transcriptomic data resulting in strain, sex, and interaction. (**E**) Gene set enrichment analyses among strain, sex, and interactions as a bubble plot with bubble size corresponding to the size of the log value (*p* < 0.05).

**Figure 4 metabolites-12-00932-f004:**
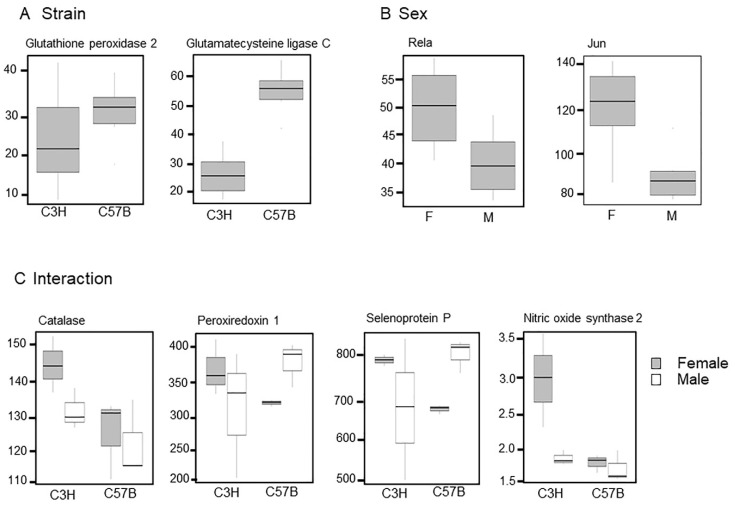
Transcripts within the inflammation, oxidant, and antioxidant pathways. Selected transcripts which differed between (**A**) strain, (**B**) sex, and (**C**) interaction were plotted.

**Figure 5 metabolites-12-00932-f005:**
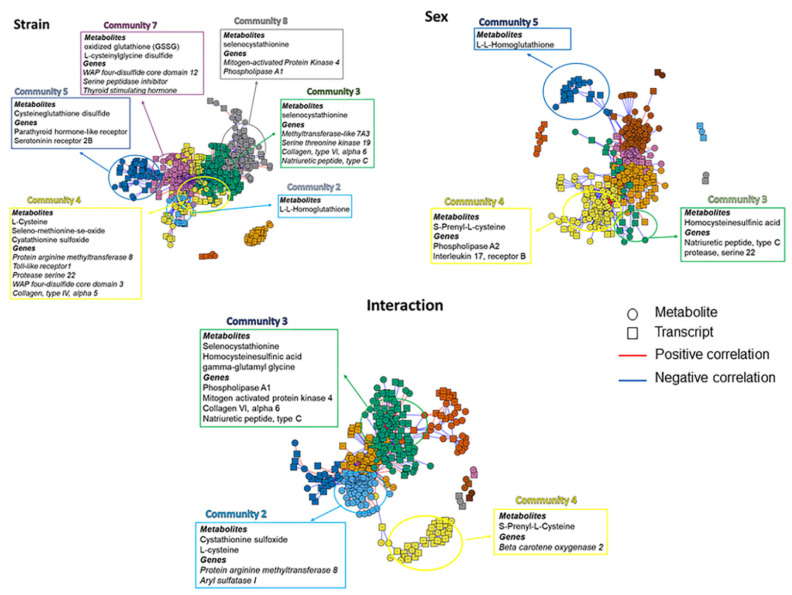
xMWAS analysis for interactions between metabolomic and transcriptomic data. Metabolites associated with oxidant/antioxidant pathways within specific communities were identified, and associated transcripts were searched using enriched data sets.

**Figure 6 metabolites-12-00932-f006:**
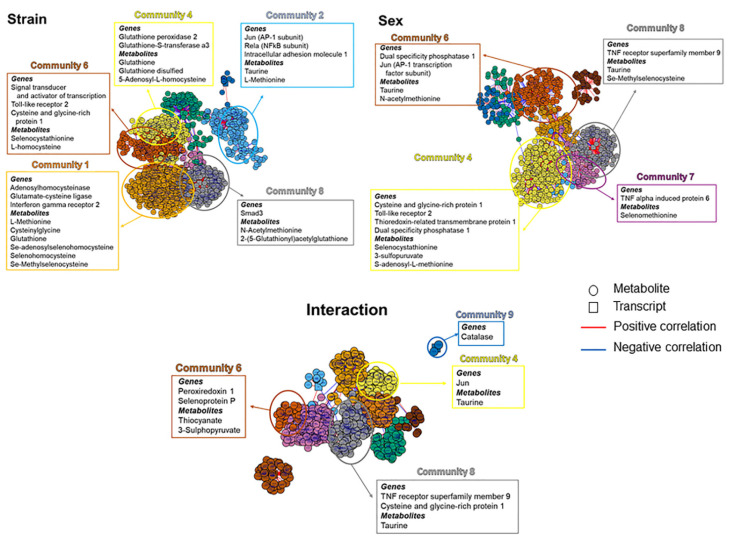
xMWAS analysis for interactions between transcriptomic and metabolomic data. Transcripts associated with inflammation and oxidant/antioxidant pathways within specific communities were identified, and associated metabolites were searched using enriched data sets.

## Data Availability

The transcriptomic data presented in this study are available in the GEO database (accession number GSE212284). The metabolomic data presented in this study are available at the NIH Common Fund’s National Metabolomics Data Repository (NMDR) website, the Metabolomics Workbench (http://dev.metabolomicsworkbench.org) where it has been assigned Project ID PR001446. The data can be accessed directly via it’s Project DOI: http://doi.org/10.21228/M8T70R. Public accessibility information: Transcriptomic data will be accessible on 27 August 27 2025 and metabolomics data will be accessible on 30 August 2023.

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
