# Peer review of "Transcriptomic-Metabolomic Profiling in Mouse Lung Tissues Reveals Sex- and Strain-Based Differences"

_metabolites, 2022, doi:10.3390/metabo12100932_

Round 1

Reviewer 1 Report

In this work “Transcriptomic-Metabolomic Profiling in Mouse Lung Tissues Reveals Sex- and Strain-Based Differences” Jolyn Fernandes et al integrated a multi-omics approach to discovery the differences of metabolites and transcripts in lung tissues amoung strain and sex. 

In general, I find the manuscript relevant to the field, and well-structured. I especially appreciated the integrated transcriptome-metabolome wide association study. However, I do feel the structure is not as well defined in the abstract, with a disproportioned amount of introduction to the topic and very little focus on the results or implications thereof. Therefore, I would suggest the authors rewrite the abstract to place additional emphasis on the results and implications. 

In the method section, the descriptions of methods were not detailed enough to allow reproducibility. Such as the setting of liquid chromatography was not supplied. How the furthers were annotated and which database was used, etc.

Author Response

Thank you for your thoughtful suggestions, we have now: 

  • Extensively editing the Abstract to remove background and focus on the results and implications.
  • We have added further details to the Methods section to better allow for reproducibility.

Reviewer 2 Report

Keywords: XMWAS? Add strain.

Introduction: line 45-55, please add information of inflammation.

Line 66-68: Further clarify the research purpose of the paper.

Fig3: sex? Gender?

Author Response

Thank you for your thoughtful review and suggestions. 

We have corrected xMWAS, added strain to the keywords, and corrected sex/gender on Figures 1 and 3.

We have added more information on inflammation and the overall purpose of this manuscript.

Reviewer 3 Report

In this study the authors explore sex- and strain-dependent effects in adult mouse lungs using high-resolution metabolomic profiling, RNA-sequencing, and an integrated assessment considering both metabolite and RNA profiles.

Although the study has the potentiality of being shared with the scientific community, I suggest a minor revision with the attempt to better support the experimental setting.

1.     The theoretical framework is scarce, they should clearly describe the scientific evidence that supports the hypothesis they have raised.

2.     The Discussion should be enriched with the existing theory. The authors should clearly describe the scientific evidence that supports their findings.

Kind regards

Author Response

Thank you for pointing the need to frame our study to support the significance of our findings.

  • We have extensively edited the Introduction to better provide a framework and previous scientific evidence to support our studies.
  • We have edited the Discussion to better focus on the information gained from these studies and its scientific value.